# Physiological skin FDG uptake: A quantitative and regional distribution assessment using PET/MRI

**Munenobu Nogami**[1]*, **Feibi Zeng**[1], **Junko Inukai**[1], **Yoshiaki Watanabe**[1], **Mizuho Nishio**[1], **Tomonori Kanda**[1], **Yoshiko R. Ueno**[1], **Keitaro Sofue**[1], **Atsushi K. Kono**[1], **Masatoshi Hori**[1], **Akihito Ohnishi**[2], **Kazuhiro Kubo**[1], **Takako Kurimoto**[3], **Takamichi Murakami**[1]

**1** Department of Radiology, Kobe University Graduate School of Medicine, Kobe, Japan, **2** Department of Radiology, Kakogawa Central City Hospital, Kakogawa, Japan, **3** GE Healthcare, Hino, Japan

* aznogami@med.kobe-u.ac.jp

## Abstract

**Data Availability Statement:** The minimal dataset is within the manuscript and Supporting Information files and additional data for analysis are available from the corresponding author.

### Purpose

To retrospectively assess the repeatability of physiological F-18 labeled fluorodeoxyglucose (FDG) uptake in the skin on positron emission tomography/magnetic resonance imaging (PET/MRI) and explore its regional distribution and relationship with sex and age.

### Methods

Out of 562 examinations with normal FDG distribution on whole-body PET/MRI, 74 repeated examinations were evaluated to assess the repeatability and regional distribution of physiological skin uptake. Furthermore, 224 examinations were evaluated to compare differences in the uptake due to sex and age. Skin segmentation on PET was performed as body-surface contouring on an MR-based attenuation correction map using an off-line reconstruction software. Bland–Altman plots were created for the repeatability assessment. Kruskal–Wallis test was performed to compare the maximum standardized uptake value (SUVmax) with regional distribution, age, and sex.

### Results

The limits of agreement for the difference in SUVmean and SUVmax of the skin were less than 30%. The highest SUVmax was observed in the face (3.09±1.04), followed by the scalp (2.07±0.53). The SUVmax in the face of boys aged 0–9 years and 10–20 years (1.33 ±0.64 and 2.05±1.00, respectively) and girls aged 0–9 years (0.98±0.38) was significantly lower than that of men aged ≥20 years and girls aged ≥10 years (p<0.001). In women, the SUVmax of the face (2.31±0.71) of ≥70-year-olds was significantly lower than that of 30–39-year-olds (3.83±0.82) (p<0.05).

Clinical data and images including personally identifiable information are not permitted to be disclosed by the ethical committee in our institution (Clinical & Translational Research Center, Kobe University Hospital) and are not accessible due to laws on the protection of personal information in our country. Please direct further data inquiries to the Clinical & Translational Research Center ethics committee (kansatsu@med.kobe-u.ac.jp).

**Funding:** Takako Kurimoto is an employee of GE Healthcare (Hino, Tokyo, Japan). The funder provided support in the form of salaries for author [TK], but did not have any additional role in the study design, data collection and analysis, decision to publish, or preparation of the manuscript. The specific roles of these authors are articulated in the 'author contributions' section. This does not alter our adherence to PLOS ONE policies on sharing data and materials.

**Competing interests:** Takako Kurimoto is an employee of GE Healthcare (Hino, Tokyo, Japan). This does not alter our adherence to PLOS ONE policies on sharing data and materials.

## Conclusion

PET/MRI enabled the quantitative analysis of skin FDG uptake with repeatability. The degree of physiological FDG uptake in the skin was the highest in the face and varied between sexes. Although attention to differences in body habitus between age groups is needed, skin FDG uptake also depended on age.

## Introduction

The skin is the largest organ of the body and plays a role in protection and sensation as well as in the synthesis and excretion of vitamin, collagen, and lipids, which require energy consumptions including glucose [1]. F-18 labeled fluorodeoxyglucose (FDG) is utilized for the assessment of several cutaneous malignancies on positron emission tomography (PET); however, its physiological uptake in the skin, reflecting the functional glucose metabolism, is yet to be evaluated. As the skin is thinner and wider than other parenchymal organs, the physiological uptake of FDG in the skin has been difficult to evaluate given the limited spatial resolution and sensitivity of PET. Moreover, skin segmentation on PET/computed tomography (CT) is challenging because of its relatively low FDG uptake and inaccurate fusion of sequential images of CT and PET due to the patient's body motion and physiological respiratory motion of the chest wall during the scan [2, 3].

Recent advances in PET detectors and novel reconstruction algorithm, including the time-of-flight (TOF) technology with high timing resolution, yielded higher spatial resolution and sensitivity than those of conventional scanners, which enabled more detailed evaluation of small structures [4, 5]. In addition, the development of an integrated PET/magnetic resonance imaging (MRI) scanner has enabled the simultaneous acquisition of PET and MRI, which allows for more precise fused images than PET/CT [6]. Therefore, simultaneously acquired PET and MRI with TOF is ideal for the evaluation of skin FDG uptake.

We hypothesized that FDG PET/MRI enables accurate and repeatable evaluation of the skin and can reveal its physiological glucose metabolism. Therefore, the purpose of the study was to evaluate the repeatability and regional distribution of physiological FDG uptake in the skin on whole-body PET/MRI and to assess the relationship between skin FDG uptake and patients' sex and age.

## Materials and methods

### Patients

The study was approved by the institutional review board (Clinical and Translational Research Center, Kobe University Graduate School of Medicine, #170032), and the requirement for informed patient consent was waived due to the retrospective nature of the study. The flowchart for the study's inclusion criteria is shown in Fig 1. We retrospectively evaluated 1348 examinations with whole-body FDG PET/MRI for oncological analysis. The maximum intensity projection PET images were reviewed by an experienced nuclear medicine physician (with 20 years of experience) to select those with a normal distribution of FDG uptake in the body according to the following criteria: 1) no apparent abnormal uptake such as in malignancy or active inflammation, 2) visually normal and maintained FDG uptake in the brain and liver, and 3) whole-body scan (from top of the head to mid-thigh) taken in the appropriate position on the bed in an arms-down position. In addition, according to the information from the electrical medical record system, patients with known dermatological diseases, including

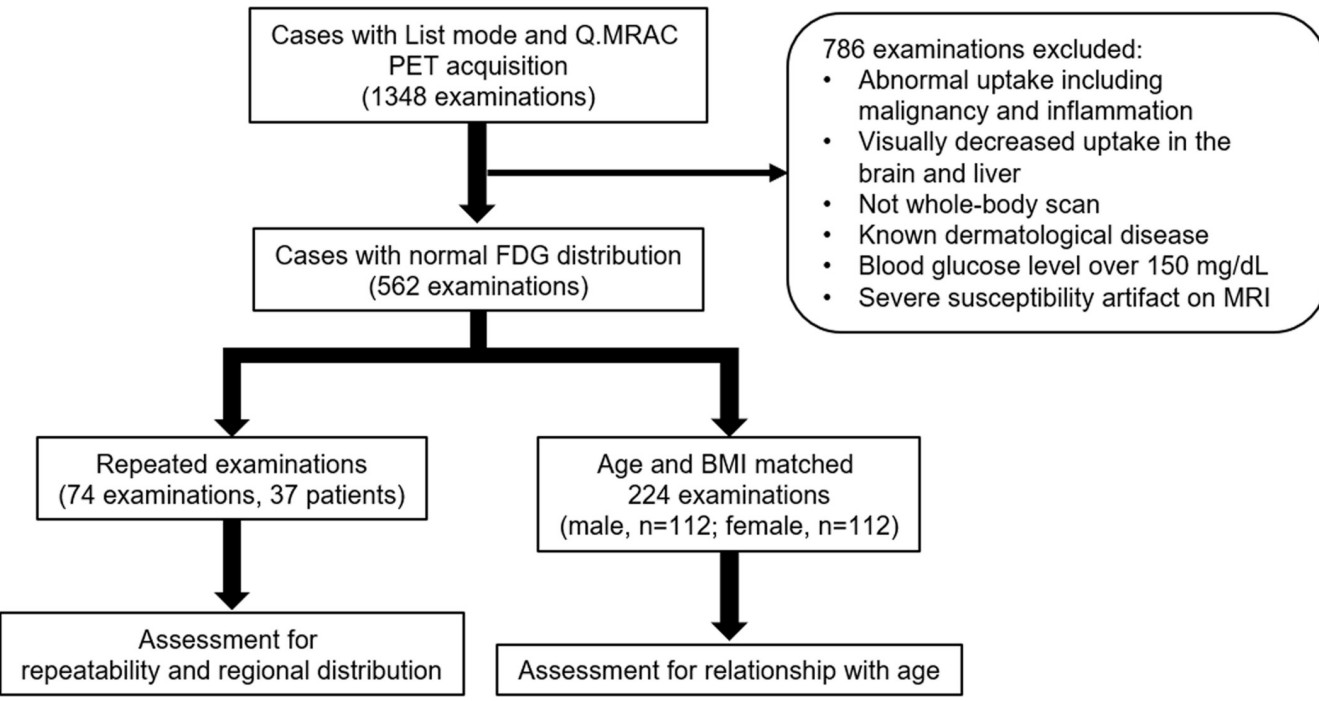

**Fig 1. Flowchart of the study inclusion criteria.**

malignant, allergic, infectious, and connective-tissue disorders, and blood glucose levels >150 mg/dL before FDG administration were excluded. Examinations with severe susceptibility artifacts on the MRI due to internal metallic objects, such as oral and femoral-head prosthesis adjacent to body surface, were also excluded. After excluding 786 examinations that met the exclusion criteria, 74 of 562 examinations (37 patients) with repeated FDG PET/MRI within one year without medical interventions were included for the assessment. Age- and body-mass-index-matched 224 of 562 examinations (male, n = 112; female, n = 112) were also selected using a statistical software and included for the assessment of the relationship of skin FDG uptake with sex and age as shown in the following statistical-analysis section. Characteristics of the included 298 examinations (261 patients) are listed in Table 1.

## F-18 FDG PET/MRI

All patients had fasted for at least 6 hours before the examinations and were administered 3.5 MBq/kg of FDG. PET was performed on an integrated PET/MRI scanner (SIGNA PET/MR, GE Healthcare, Waukesha, Wisconsin, the United States) at 3.0 T in the magnetic field strength. For whole-body acquisition, four to six bed positions were required for the PET scan in the arms-down position to cover the top of the head to the mid-thigh, with an axial field of view (FOV) of 25 cm. A PET scan was recorded in the list mode and performed for 2.5 min for each of the bed positions, except for 5.0 min in the thoracic bed position for the following respiratory-gated reconstruction. Magnetic resonance attenuation correction (MRAC) scans were simultaneously performed with PET using 2-point Dixon three-dimensional volumetric interpolated fast spoiled gradient echo sequence (LAVA-Flex) in combination with respiratory-gated MRAC (Q.MRAC) in the thoracic bed position. PET was reconstructed by TOF ordered subset expectation maximization (TOF-OSEM) with a transaxial FOV of 600 mm, $192 \times 192$ matrix (matrix size, $3.125 \times 3.125 \times 2.809$ mm$^3$), 2 iterations, 16 subsets, and a

**Table 1. Patient characteristics.**

| Assessment for repeatability and regional distribution (74 examinations, 37 patients) | | | | | |
|---|---|---|---|---|---|
| | 1st examination | | 2nd examination | | p value for BMI |
| | Number | BMI | Number | BMI | |
| | 37 | 22.1 (19.9–23.6) | 37 | 21.6 (18.8–23.0) | 0.378 |
| Assessment for relationship with age (224 examinations, 224 patients) | | | | | |
| Age group | Male | | Female | | p value for BMI |
| | Number | BMI | Number | BMI | |
| 0–9 | 10 | 14.9 (13.9–15.0) | 7 | 15.3 (15.0–15.7) | 0.063 |
| 10–19 | 15 | 17.7 (13.9–22.2) | 14 | 18.0 (13.3–23.0) | 0.930 |
| 20–29 | 8 | 23.5 (20.2–27.6) | 8 | 20.1 (19.0–21.7) | 0.093 |
| 30–39 | 13 | 23.9 (19.7–27.0) | 13 | 23.9 (19.3–25.0) | 0.489 |
| 40–49 | 15 | 21.5 (19.1–22.5) | 15 | 24.3 (20.4–24.2) | 0.120 |
| 50–59 | 16 | 21.3 (19.3–23.4) | 20 | 22.0 (19.7–23.3) | 0.750 |
| 60–69 | 14 | 22.8 (21.0–25.5) | 15 | 21.0 (19.1–22.3) | 0.089 |
| 70–79 | 21 | 23.8 (21.2–26.5) | 20 | 21.9 (18.1–26.7) | 0.085 |
| Overall | 112 | 21.4 (18.7–23.8) | 112 | 21.4 (18.4–23.0) | 0.435 |

Numbers for the BMI is median and interquartile range (parenthesis).

Gaussian filter of 4.0 mm with a point-spread function. For the thoracic bed position, the PET reconstruction was performed with the quiescent period gating (Q.Static) with offset/acquisition windows of 30/50%, respectively.

## Image segmentation

To extract a skin region in the whole body on PET, simultaneously acquired MRAC images were utilized. Details of the segmentation method are shown in Fig 2. First, MRAC images

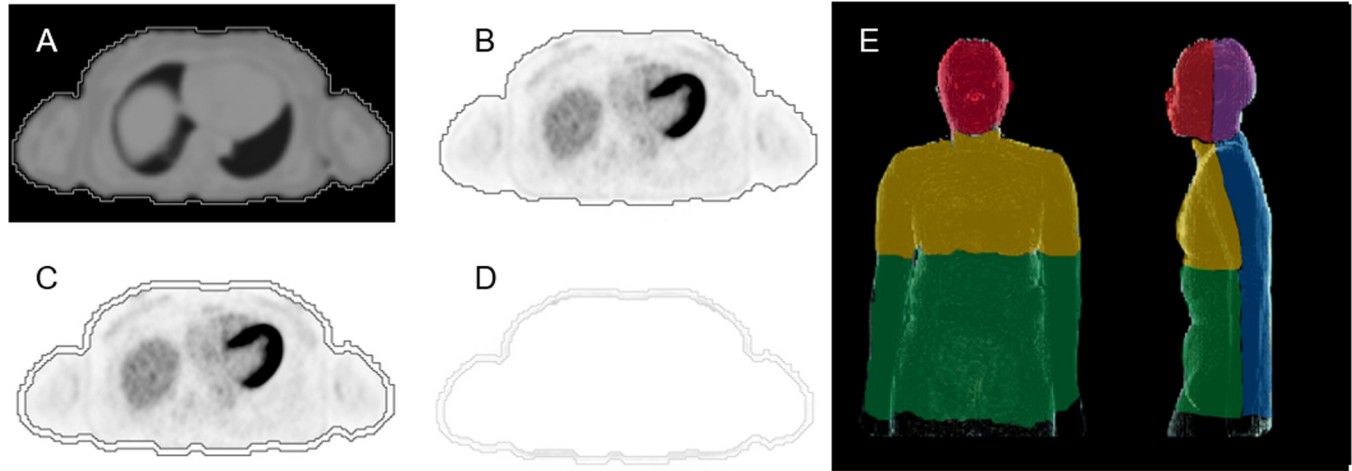

**Fig 2. Segmentation procedure for VOI creation of the skin.** From the PIFA DICOM images, VOI for body contour was generated by segmentation on the PIFA images with a threshold value (A). The VOI was directory copied to the simultaneously acquired PET images (B). Three voxels were extracted from the body contour to the inside of the body (C). The VOI for the skin was then generated by subtracting C from B (D). The VOI of the skin was separately assessed in the face (red), scalp (purple), chest (yellow), abdomen (green), and back (blue) (E). The borders of the regions were defined as the vertical line from the external ear canal for head, the middle of the arms for the trunks, and the lower end of the 12th thoracic vertebra. VOI, volume-of-interest; PIFA, positron emission tomography images for attenuation; DICOM, Digital Imaging and Communication in Medicine.

without attenuation information of MR beds and coils (in-vivo PET images for attenuation [PIFA]) were generated from MATLAB-based off-line PET reconstruction tool (PETtoolbox and Duetto, GE Healthcare) using the original list files of PET and LAVA-Flex for MRAC [7, 8]. The in-vivo PIFA was created by the same FOV of the reconstructed PET images and generated in the Digital Imaging and Communication in Medicine (DICOM) format. Second, in-vivo PIFA DICOM was transferred to a commercially available workstation (Advantage Workstation 4.7, GE Healthcare) and processed by an image segmentation tool. To generate the volume-of-interest (VOI) for the body contour, the body surface of the in-vivo PIFA was automatically traced with a threshold. Body contour segmentation was easily performed using a threshold alone because the value of the background signal intensity outside of the body contour was set at null in the process of in-vivo PIFA generation from MRI. Although the skin's thickness is anatomically <5 mm [9], margins were required for creating VOIs greater than the exact skin thickness to robustly include tracer uptake in the skin. In addition, the margins were adjusted depending on the patients' body size so as not to include the physiological uptake of the adjacent organs into the VOI. Therefore, three voxels (9.375 mm) were extracted from the body contour to the inside of the body to create the VOIs for the skin.

## Image analysis

All image analyses were performed using a commercially available workstation (Advantage Workstation 4.7, GE Healthcare). VOIs of the skin of each patient were copied to attenuation-corrected PET images without misregistration owing to the simultaneous and respiratory-gated acquisition of PET scan and MRAC. On the PET images, the copied VOIs were carefully reviewed by an experienced nuclear medicine physician so as not to include adjacent physiological FDG uptakes such as those of the brain, lacrimal and salivary glands, muscle including external ocular muscle, liver, and urinary system. Manual adjustment of the VOIs were made on the workstation when FDG uptake of other organs into the VOIs were noted. Thereafter, mean standardized uptake values (SUVmean) and maximum SUV (SUVmax) of the overall skin were measured from the VOI placement and separated into five regions in the body, namely, face, scalp, chest, abdomen, and back regions, to assess the repeatability and regional distribution of skin FDG uptake (Fig 2). SUVs were calculated as follows:

$$SUV = \text{tissue activity } [Bq/mL]/(\text{injected dose } [Bq]/\text{body weight } [g])$$

Additionally, the SUVmean and SUVmax in the liver as a reference SUV in each examination was measured by averaging the values in three identical cubic voxels of 15 mm placed separately in the normal liver uptake.

## Statistical analysis

To select age- and body-mass-index-matched examinations for the assessment of the relationship of skin FDG uptake with sex and age, the Wilcoxon signed-rank test was performed to confirm that the 95% confidence interval (CI) of the difference and associated p-value were negligible and nonsignificant between the sexes in each group divided by every 10 years, from 0–80-year-old. To evaluate the repeatability of the skin FDG uptake, the Bland–Altman plot was created to assess the mean difference and limits of agreements of the overall skin SUVmean and SUVmax and in each of the regions as well as those in the liver. The coefficient of repeatability and 95% CI for the Bland–Altman plot were calculated according to Bland and Altman [10] and Barnhart and Barborial [11], respectively. To investigate the regional distribution of skin FDG uptake, the Kruskal–Wallis test followed by the pairwise comparison was performed to assess the difference in SUVmax between regions with unequal variances according

to Conover [12]. To evaluate the relationship of skin SUV with sex and age, Kruskal–Wallis test was used to statistically compare the skin SUVmax among the groups divided based on sex and age. Variables were presented as mean ± standard deviation (SD). All statistical analyses were performed using MedCalc, version 19.1.6 (MedCalc Software Ltd, Ostend, Belgium). P-values $<0.05$ were considered significant.

## Results

Manual adjustment of the VOIs was necessary for 69 of the 298 examinations (23.2%) due to physiological uptake in the brain (n = 56), salivary gland (n = 47), muscle (n = 32), liver (n = 28), and urinary excretion (n = 62) (multiple adjustments were necessary within the same examination).

### Repeatability of skin FDG uptake

The mean interval of time between repeated examinations was 125.6 days (range, 35–239; median, 126.5). The percent mean difference and lower and upper limits of agreements of the SUVmean (and 95% CI) in the overall skin region were -0.918 (-4.828 to 2.991), -23.899 (-30.641 to -17.157), and 22.063 (15.321 to 28.804), respectively, and those for the liver were -1.250 (-6.346 to 3.845), -31.204 (-39.991 to 22.416), and 28.703 (19.916 to 37.491), respectively. The percent mean difference and lower and upper limits of agreements of the SUVmax (and 95% CI) in the overall skin region were –0.5649 (-5.011 to 3.881), -26.700 (-34.367 to -19.033), and 25.571 (17.903 to 33.238), respectively, and those for the liver were -0.934 (-5.642 to 3.775), -28.613 (-36.733 to -20.493), and 26.746 (18.626 to 34.866), respectively (Fig 3). The coefficient of repeatability for the SUVmean (95% CI) in the overall skin region and liver were 0.052 (0.042 to 0.067) and 0.642 (0.523 to 0.830), respectively, and that for the SUVmax (95% CI) in the overall skin region and liver were 0.871 (0.711 to 1.127) and 0.749 (0.611 to 0.969), respectively. The repeatability for the SUVmean in each region is shown in S1 Fig and S1 Table.

### Regional distribution of skin FDG uptake

The highest skin SUVmax was observed in the face (3.09±1.04), followed by the scalp (2.07±0.54), back (1.47±0.34), chest (1.33±0.34), and abdomen (1.24±0.49) (Figs 4 and 5, and S2 Table). The SUVmax of the face was significantly higher than that of the other regions (p<0.0001), and that of the scalp was significantly higher than that of the back, chest, and abdomen (p<0.0001). The SUVmax of the back was significantly higher than that of the abdomen (p = 0.0017), whereas no significant difference in the SUVmax was found between the chest and abdomen (p = 0.3457).

### Relationship of skin SUVmax with sex and age

The highest skin SUVmax was observed in the face region in all examinations (n = 224). The SUVmax of the face in male patients (3.45±1.48) was significantly higher than that in female patients (2.87±1.13) (p = 0.0012). In male patients, the highest SUVmax of teens was significantly higher than that of those aged between 0 and 9 years (p<0.001) and significantly lower than that of those of the other age groups (p<0.005). In female patients, the highest SUVmax in 0–9-year-olds was significantly lower than that in all other age groups (p<0.001), whereas the value in individuals in their 70s was significantly lower than that in individuals in their 30s (p<0.001) (Figs 6 and 7, and S3 Table).

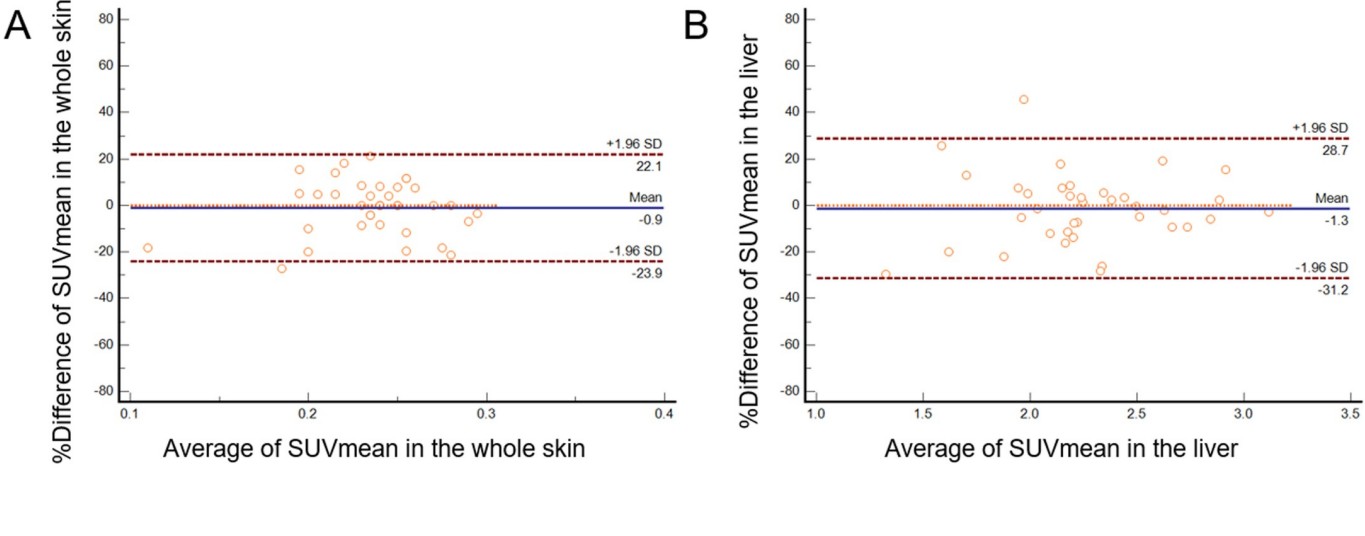

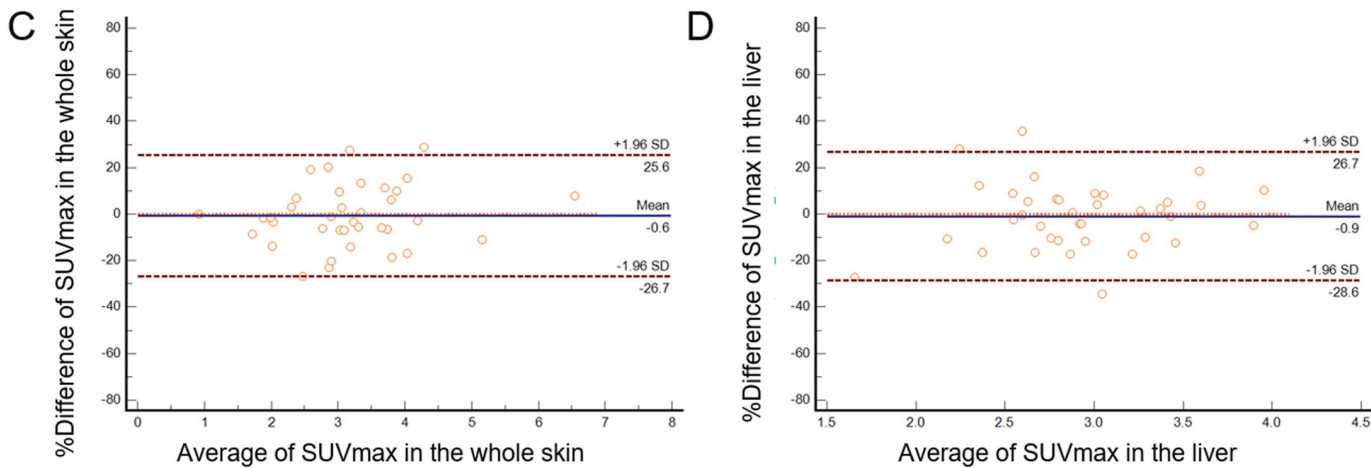

**Fig 3. The Bland–Altman plot representing the repeatability of the SUVmean and SUVmax average.** Average SUVmean and SUVmax in the overall skin region (A and C, respectively) and in the liver (B and D, respectively) (37 patients, 74 examinations). The limits of agreements of percent difference of SUVmean and SUVmax were within 30% in the overall skin region and smaller than those in the liver. SUVmean, mean standardized uptake value; SUVmax, maximum SUV standardized uptake value.

## Discussion

Our results demonstrated the repeatability of physiological FDG uptake in the skin by simultaneously acquired PET/MRI. Quantitative assessment of the skin SUV revealed significant differences in the physiological skin FDG uptake between skin regions, sexes, and ages.

Formerly, cutaneous FDG uptake was difficult to evaluate using attenuation-corrected images and was commonly appreciated using non-attenuation corrected images [13, 14]. However, accurate attenuation correction is necessary to quantitatively assess the tracer uptake on PET, but this is challenging for skin evaluation due to its anatomically thin structure. Although theoretically, CT is more advantageous for PET attenuation correction than MRI owing to the linear relationship between X-ray and γ-ray attenuation information, misregistration due to sequential acquisition of CT and PET and/or patient movements, including respiratory motion, during scan cannot be avoided when considering the configuration and imaging procedure of PET/CT [15]. Although MR-based attenuation correction suffers from a

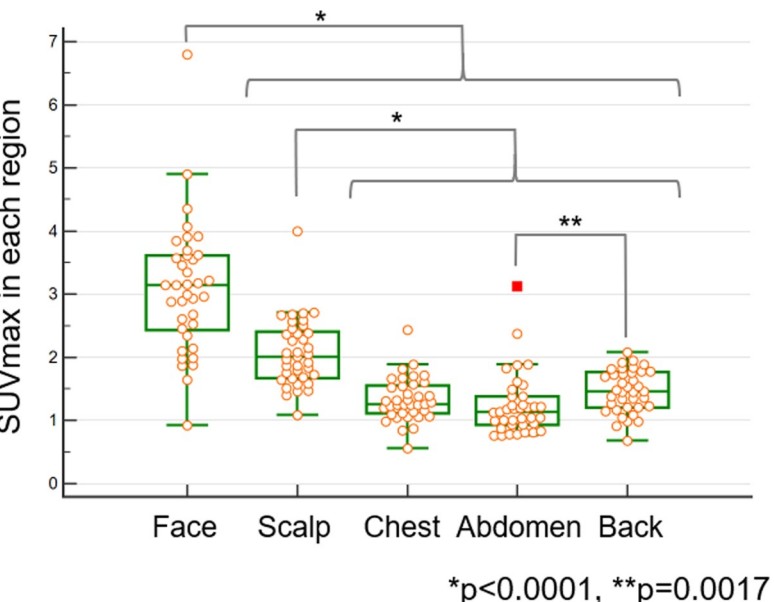

**Fig 4. The box and whisker plot representing the SUVmax in each skin region (37 examinations).** SUVmax in the face was significantly higher than that in the other regions (p<0.0001), and that in the scalp was significantly higher than that in the back, chest, and abdomen (p<0.0001). The SUVmax in the back was significantly higher than that in the abdomen (p = 0.0017), whereas there was no significant difference in SUVmax between the chest and abdomen (p = 0.3457). SUVmean, mean standardized uptake value; SUVmax, maximum SUV standardized uptake value.

lack of attenuation information of the bones, MRAC map is more precise than a CT-based attenuation correction (CTAC) map for registration with non-attenuation corrected PET generated from simultaneous- and respiratory-triggered acquisition [16, 17]. In addition,

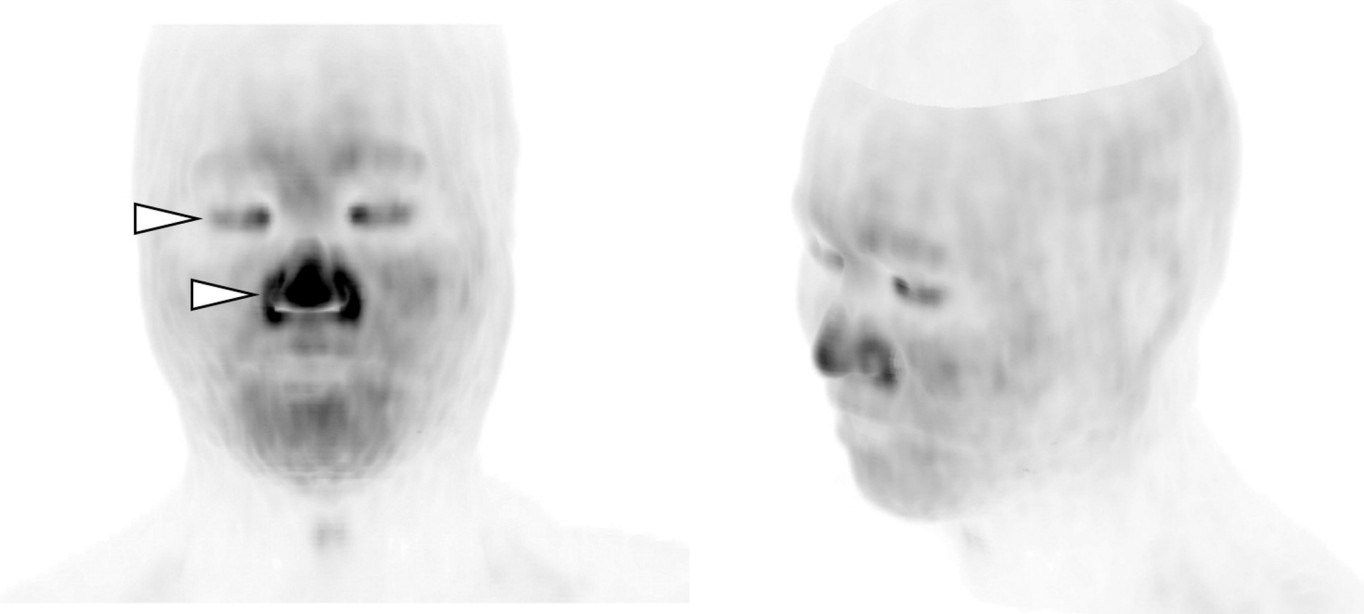

**Fig 5. Weighted MIP images of skin FDG uptake in the face.** Separately scanned: Emission scan, 9 minutes; 392 × 392 matrix, BSREM (block sequential regularized expectation maximization, Q.Clear, β = 350) reconstruction. Note the increased uptake in the eyelids and tip of the nose (arrow heads). MIP, maximum intensity projection.

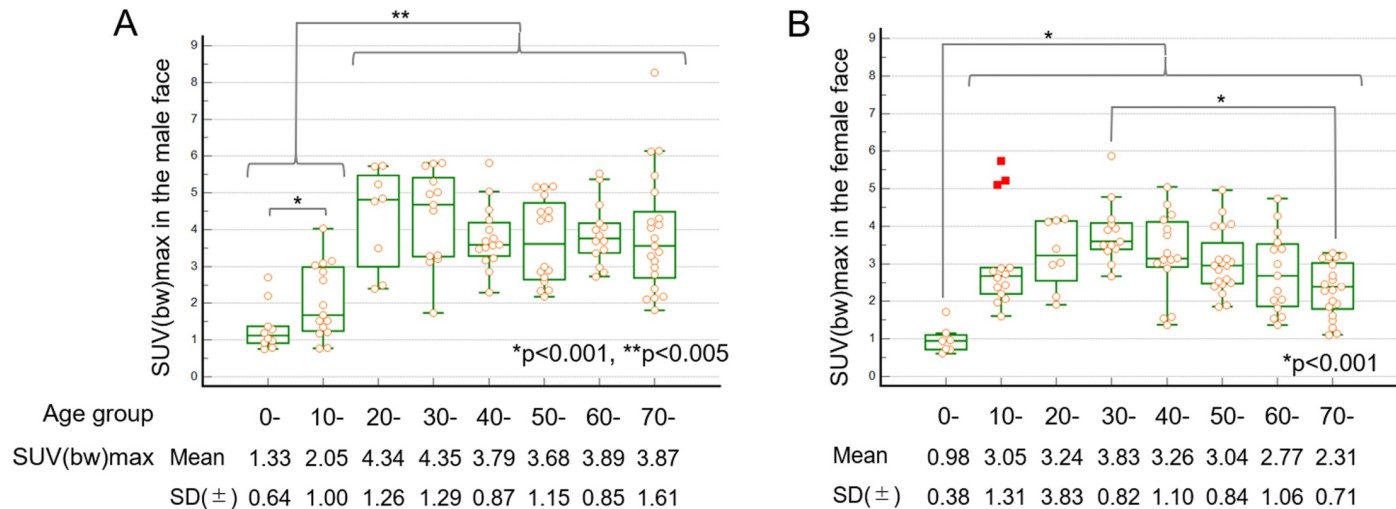

**Fig 6. The relationship between skin SUVmax in the face and age group.** The box and whisker plot representing the SUVmax in male (n = 112) (A) and female (n = 112) (B) patients. In male patients, the skin SUVmax of teenage patients was significantly higher than that of boys 0–9-year-old (p<0.001) and significantly lower than that of individuals in the other age groups (p<0.005) (A). In female patients, the SUVmax of 0–9-year-olds was significantly lower than that of all the other age groups (p<0.001), and the SUVmax of those in their 70s was significantly lower than that of those in their 30s (B). SUVmean, mean standardized uptake value; SUVmax, maximum standardized uptake value.

generated PIFA is free from CT-derived artifacts such as beam-hardening artifacts and is more suitable for segmentation of the body contour than CTAC map because the background is set as a null value and is easy for segmentation with a fixed threshold value.

Although SUVmax is a simple measure and has been widely utilized for the analysis of PET images [18], it is important to realize that many factors could affect the accuracy of SUV

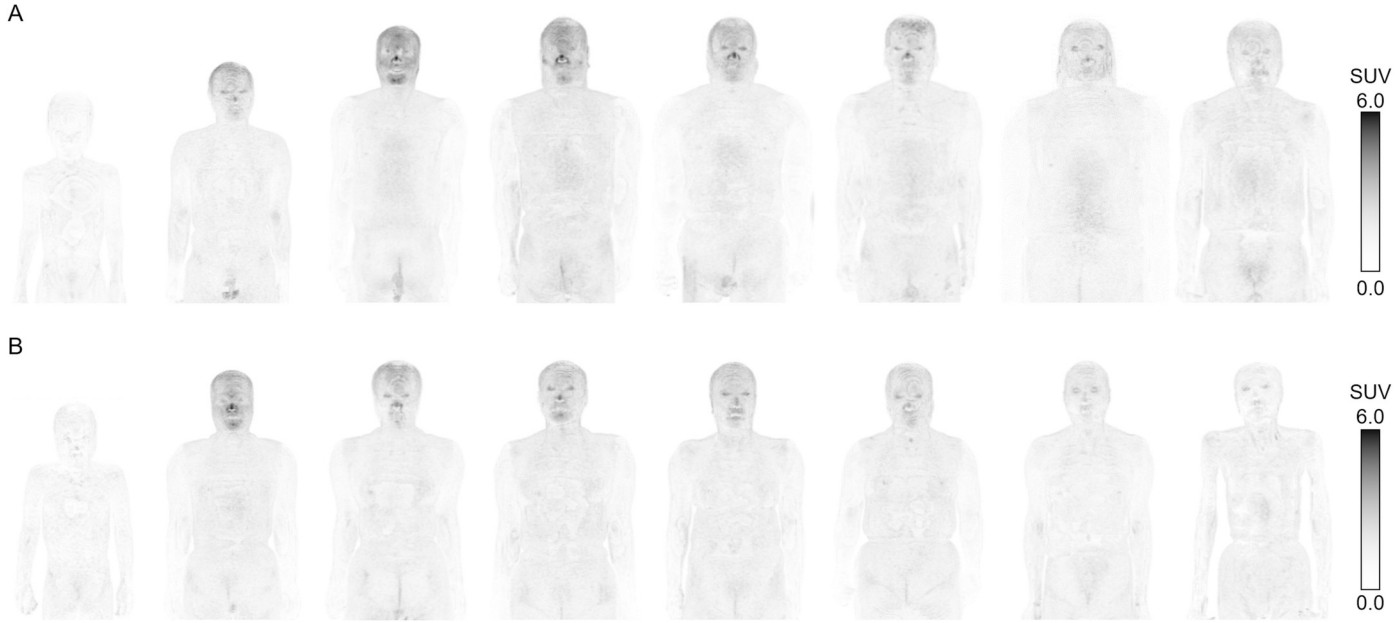

**Fig 7. Representative MIP images showing regional distributions of skin FDG uptake and their differences between sexes.** (A) male and (B) female patients, and age groups (left to right, age group of 0–9, 10–19, 20–29, 30–39, 40–49, 50–59, 60–69, and 70–79 years, respectively). The uptake was highest in the face in both sexes and increased from puberty. The uptake was maintained until the age of 70 years in male patients but decreased after the menopausal age in female patients. MIP, maximum intensity projection. FDG, fluorodeoxyglucose.

measurements including biological and technical factors[19]. Because the method of normalization of SUV [the patients' body weight (bw)] could affect the quantitative value and its relationship with age, additional assessments using other SUV metrics, such as normalization by lean body mass (lbm) or body surface area (bsa), were performed and showed no difference in the tendency regarding the relationship between SUVmax and age established in this study (S2 and S3 Figs). However, caution is required when comparing SUVs between different age groups because a certain difference in body mass index is present in pediatric patients. As shown in (S4 Fig), normal liver SUVs were also significantly lower in pediatrics than adult patients in both sexes, suggesting that the differences in skin SUV in the pediatrics were not related to absolute FDG uptake at their age, but rather the biased normalization procedure for SUV based on their body habitus.

The mechanism of skin FDG uptake has not yet been elucidated but could be explained by its distribution in the body and/or sex- or age-related differences in function. The skin consists of the epidermis and corium, which are 30 μm to 4 mm and 300 μm to 4 mm in thickness, respectively [9]. The skin of the back is the thickest in the body; however, our results indicated that SUVmax in the back was significantly lower than that in the face and scalp, indicating that another possible cause for FDG uptake other than the skin thickness could be assumed. Another possible reason for skin FDG uptake is the blood pool in the capillary blood. However, the SUVmax in the face, especially in the nose, was higher than that in the large vessels and even higher than that in the liver in 58% (131 of 224 examinations) of the patients in our study (S5 Table), suggesting that the skin uptake cannot be explained by the blood pool alone. As FDG is a glucose analog, high accumulation in the skin could occur where high expression of the glucose transporter and/or glucose phosphorylating enzyme hexokinase is found [20]. With respect to the spatial resolution of PET, high cellular density is also required to visualize high uptake on the images. In the normal skin, several structures that accumulate FDG were expected, such as the muscles, peripheral nerves, and glands.

From the results of our evaluation for regional distribution of FDG uptake, the face had the highest uptake in both sexes, followed by the scalp. When the difference between the age groups was assessed, uptake in the face increased at puberty and was maintained until the age of 70 years in male patients but decreased after the menopausal age in female patients. Although assessment of SUV in pediatrics requires attention to the difference in body mass index as discussed above, these findings were concordant to functions of the sebaceous glands in the skin with respect to the distribution in the body and sex- and age- differences regulated by an endocrine factor, including androgen [21–24]. Major glands in the skin are composed of sweat and sebaceous glands; however, a greater number of sweat glands are present in the hairy skin regions as well in the palms and soles [25]. Although the lower extremities were not included in the scan in our study, no apparent FDG uptake was observed in the palms, which is discordant to the distribution of the sweat gland. Sebaceous gland, on the contrary, is located in a hair follicle forming a pilosebaceous unit or apparatus, such as in the scalp [26, 27], and in areas independent of hair follicles (free sebaceous glands), such as the eyelids, edge of the lip of the mouth, but not in the palms. In addition, large sebaceous glands are found on the tip of the nose, nipple, and the auricle of the ear [22, 28], which corresponds to the distribution of FDG uptake in our study. In this study, from a clinical and ethical standpoint, histopathological verification of this hypothesis was not permitted, and further research to clarify the mechanism of physiological FDG uptake in the skin is warranted; however, our result suggested that the normal FDG uptake in the skin was related to the distribution and function of the sebaceous glands.

Our study has several limitations. First, the area below the mid-thigh could not be evaluated due to the limited scan range from the top of the head to the mid-thigh level. In addition, the

upper arms of patients of a large size could have been outside the FOV on MRI and were filled with a fixed attenuation value on PIFA (truncation completion), resulting in the possibility of an inaccurate SUV measurement in those regions. Additionally, the skin inside the upper arm, including the axillary area, could not be evaluated using the segmentation method because the scans were performed with the arms down. The axillary region is a common region of sebaceous gland distribution and is preferably assessed with the arms raised; however, the limited gantry space of PET/MRI hampers such position, and further assessment is mandatory for the quantitative evaluation of the axillary region. Second, although only those without any interventions for known malignancies were selected, all patients had a history of malignancy and were not healthy volunteers, resulting in possible biased results that could be drawn from the study population; however, exclusion of patients with an active state in malignancy would be sufficient for evaluation of physiological FDG uptake of the skin. Third, the presented method for skin segmentation on PET based on the simultaneously acquired MRAC yielded SUV measurements in the surface area of the body; however, precise measurements of the skin FDG uptake is still difficult because the thin and vague structure cannot be accurately visualized and segmented on MRI alone, resulting in influences of individual differences in the skin and subcutaneous-tissue thickness, and spillover effects by adjacent high FDG uptake in the other organs, such as the brain and urinary system, cannot to be ignored. Although the segmented VOIs were carefully double-checked and fine-adjusted to measure the skin FDG uptake, further investigations regarding accurate and robust segmentation of the skin are warranted.

## Conclusion

PET/MRI enabled the quantitative analysis of skin FDG uptake with repeatability. The degree of physiological skin FDG uptake was the highest in the face and varied depending on sex in a manner similar to the distribution and function of the sebaceous gland. Although attention to differences in body habitus between age groups is needed, skin FDG uptake also depended on age.

## Supporting information

**S1 Fig. Bland Altman plots for difference of SUVmean between the repeated examinations in each region.** The graphs show face (A), scalp (B), chest (C), abdomen (D), and back (E) regions.
(DOCX)

**S2 Fig. Relationship between skin SUV(lbm)max in the face and age group.**
(DOCX)

**S3 Fig. Relationship between skin SUV(bsa)max in the face and age group.**
(DOCX)

**S4 Fig. Relationship between liver SUV(bw)max and age group.**
(DOCX)

**S1 Table. Repeatability of skin SUVmean in each region (n = 37).**
(DOCX)

**S2 Table. SUVmax of each region (n = 37).**
(DOCX)

**S3 Table. SUVmax in the face in each sex and age group (n = 224).**
(DOCX)

**S4 Table. Raw-data table of patients for the repeatability assessment (n = 37).**
(DOCX)

**S5 Table. Raw-data table of patients for the relationship with age (n = 224).**
(DOCX)

## Acknowledgments

We would like to thank Editage (www.editage.com) for English language editing.

## Author Contributions

**Conceptualization:** Munenobu Nogami, Akihito Ohnishi.

**Data curation:** Munenobu Nogami, Feibi Zeng, Junko Inukai, Tomonori Kanda, Kazuhiro Kubo.

**Formal analysis:** Munenobu Nogami.

**Funding acquisition:** Takamichi Murakami.

**Investigation:** Yoshiko R. Ueno.

**Methodology:** Feibi Zeng, Junko Inukai, Mizuho Nishio, Yoshiko R. Ueno, Keitaro Sofue.

**Project administration:** Atsushi K. Kono, Takamichi Murakami.

**Resources:** Yoshiaki Watanabe, Mizuho Nishio, Akihito Ohnishi.

**Software:** Kazuhiro Kubo, Takako Kurimoto.

**Supervision:** Masatoshi Hori, Takamichi Murakami.

**Validation:** Mizuho Nishio, Atsushi K. Kono.

**Visualization:** Yoshiaki Watanabe, Tomonori Kanda.

**Writing – original draft:** Munenobu Nogami.

**Writing – review & editing:** Mizuho Nishio, Keitaro Sofue, Masatoshi Hori, Akihito Ohnishi.

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
