## [Decision Letter · Decision Letter 0]

7 Dec 2020

PONE-D-20-29259

Physiological skin FDG uptake: A quantitative and regional distribution assessment using PET/MRI with a silicon photomultiplier

PLOS ONE

Dear Dr. Nogami,

Thank you for submitting your manuscript to PLOS ONE. After careful consideration, we feel that it has merit but does not fully meet PLOS ONE’s publication criteria as it currently stands. Therefore, we invite you to submit a revised version of the manuscript that addresses the points raised during the review process.

We look forward to receiving your revised manuscript.

Kind regards,

Matteo Bauckneht, MD, PhD

Academic Editor

PLOS ONE

Journal Requirements:

"Takako Kurimoto is an employee of GE Healthcare (Hino, Tokyo, Japan). "

We note that one or more of the authors are employed by a commercial company: GE Healthcare.

3.1. Please provide an amended Funding Statement declaring this commercial affiliation, as well as a statement regarding the Role of Funders in your study. If the funding organization did not play a role in the study design, data collection and analysis, decision to publish, or preparation of the manuscript and only provided financial support in the form of authors' salaries and/or research materials, please review your statements relating to the author contributions, and ensure you have specifically and accurately indicated the role(s) that these authors had in your study. You can update author roles in the Author Contributions section of the online submission form.

3.2. Please also provide an updated Competing Interests Statement declaring this commercial affiliation along with any other relevant declarations relating to employment, consultancy, patents, products in development, or marketed products, etc.  

Reviewers' comments:

Reviewer's Responses to Questions

**Comments to the Author**

1. Is the manuscript technically sound, and do the data support the conclusions?

Reviewer #1: Partly

Reviewer #2: Partly

2. Has the statistical analysis been performed appropriately and rigorously? 

Reviewer #1: Yes

Reviewer #2: Yes

3. Have the authors made all data underlying the findings in their manuscript fully available?

Reviewer #1: No

Reviewer #2: Yes

4. Is the manuscript presented in an intelligible fashion and written in standard English?

Reviewer #1: Yes

Reviewer #2: Yes

5. Review Comments to the Author

Reviewer #1: “Physiological skin FDG uptake: A quantitative and regional distribution assessment using PET/MRI with a silicon photomultiplier” is a basic science paper seeking to elucidate the magnitude, distribution, age and gender dependence, and reproducibility of the uptake of FDG in human skin. It bases this assessment on measurements taken from whole-body 18F-FDG PET/MR studies involving a final cohort (after exclusions and patient matching) of 261 patients, 37 of which were used to make comparisons between a pair of exams (thus 298 exams altogether – I believe, although because of the convoluted description it is hard to be certain). The skin was divided into 5 regions and SUVmax and SUVmean measurements were made for each. Segmentation of the skin was based on the MRAC images. SUVmax and SUVmean measurements were also taken of the liver.

For the most part the technical aspects of this study appear to have been performed well and carefully with a lot of attention devoted to removing confounds from underlying organs etc. I do somewhat question the decision to use 2-voxel thick VOIs for the <10-year olds but 3-voxel thick sections for the rest. I understand that this was done because of the differences in skin thickness, but this is a rather dramatic difference at a fair arbitrary cut off (age 10) and could easily lead to bias in the age dependent assessments. Of course, using the same VOI thickness for all could (likely would) also incur bias, but arguably this bias would be more predictable and perhaps even compensated for using a partial volume correction based upon an age dependent function of skin thickness.

Partial volume issues in general are a major confound in this study. As noted in the manuscript, skin thickness is far from uniform across the body, as are the relative thicknesses of the dermis, epidermis and makeup of the other skin components. PET resolution and accuracy at this scale are simply not sufficient to make realistic absolute measurements of structures this small, especially at the extreme surface of the body.

My biggest concern about this work, however, is how it treats SUV measures as if they were quantitative measures of FDG transport. I assume the SUV metric used here is the one normalized by patient body-weight (the formula is not described but my concern would apply equally had SUV normalized by lean-body mass or body surface-area etc. been used) and this is by no means a perfect normalizer. It is quite possible (likely even) that much of the FDG uptake age dependence described is actually due to the choice of normalizer. As a check, it would be interesting to see what the liver uptakes dependence on age was.

In spite of these deficits I do think that this work could contain information useful to the research community. For it to be so, however, it would be better if the authors simply made as much of the raw data available as possible (e.g. a table of heights, weights, ages, SUVmeans, SUVmaxes etc. for each subject) and using VOIs of the same thickness for all subjects.

And one last minor note. Much is made (in the title, abstract and text) of the fact that the PET camera used contain silicon photomultipliers, but there’s no clear reason to think that these per se had any real influence on this study. Yes, PET/MR was likely had advantages over PET/CT but not because the latter in some cases uses PMTs.

Reviewer #2: While the hypothesis, methodology, and reported results appear appropriate, cogent, and logical, there are several items lacking that should be addressed in the Discussion (or with further evaluation).

Discuss the possibility that the observed results are due to backscatter from bony surfaces in face, skull, sternum (not present in abdomen, etc.). Any literature in that area to review?

Discuss the possibility that the observed results may be in part explained by the lack of self shielding in the higher noted uptake areas.

Discuss the possibility that the observed results may be the result of uptake in the blood pool (higher capillary blood at the surface in the noted higher uptake regions), in the eyes themselves (or salivary glands), in the nasopharangeal regions.

Are any of these potentials a limitation of the methodology?

6. PLOS authors have the option to publish the peer review history of their article (what does this mean?). If published, this will include your full peer review and any attached files.

Reviewer #1: No

Reviewer #2: No

---

## [Author Response · Author response to Decision Letter 0]

4 Feb 2021

We thank the reviewers for the evaluation of our study and the editor for giving us the chance of resubmitting our manuscript. We are happy to address all comments point by point. We have made revisions that have been marked in colored fonts for ease of identification in the revised version of our manuscript.

Reviewer #1: 

1. Comments: 298 exams altogether – I believe, although because of the convoluted description on it is hard to be certain.

Response: We apologize for any confusion our description may have caused. As you mentioned, the total number of assessed examinations was 298. The heading of Table 1 was modified to clearly show the numbers of assessed examinations per category.

2. Comments: I do somewhat question on the decision to use 2‐voxel thick VOIs for the <10‐year olds but 3‐voxel thick sections for the rest. …this bias would be more predictable and perhaps even compensated for using a partial volume correction based upon an age dependent function of skin thickness.

Response: Thank you for your suggestions. As you pointed out, differences in VOI thickness were due to age-dependent differences in skin thickness. The 9.375 mm thickness of 3 voxels was too large for the segmentation of the thin skin in children, especially when VOIs were adjacent to organs such as the brain with high physiological uptake. This resulted in a lot of manual adjustments to exclude voxels outside of the skin structure (as shown in the following figure).

However, we agree that this difference in the definition of VOIs may lead to unnecessary bias in this study. Therefore, we have performed additional image segmentation using the same VOI thickness (3 voxels) for all examinations. Accordingly, more manual adjustments were necessary for patients under the age of 10 years; however, the statistical analysis showed no difference in the results from the previous assessment with different VOI segmentations. The modified number of the manual adjustments are shown in the results section (Page 11, lines 174 to 176). A slight difference in SUVmax in the face was found in patients under 10 years of age, but the statistical results were not different in comparison to the previous assessment (Figure 6).

3. Comments: I assume the SUV metric used here is the one normalized by patient body‐weight (the formula is not described but my concern would apply equally had SUV normalized by lean‐body mass or body surface‐area etc. been used) and this is by no means a perfect normalizer. It is quite possible (likely even) that much of the FDG uptake age dependence described is actually due to the choice of normalizer. As a check, it would be interesting to see what the liver uptakes dependence on age was.

Response: Thank you very much for your extremely valuable comments. The SUV calculation was normalized to the patients’ body weight, as you mentioned. We added a description regarding the definition of the SUV calculation to the Image analysis section (Page 10, line 151). Furthermore, we do agree that the selection of a normalizer (body weight [bw], lean body mass [lbm], and body surface area [bsa]) could have an impact on the statistical difference in SUVs among age groups. Therefore, we calculated SUVmax with lbm and bsa as well to assess differences in SUVs among age groups and found that both SUV(lbm)max and SUV(bsa)max showed no difference in relation to age compared to SUV(bw)max, suggesting the age-dependency of the skin uptake was not due to the choice of the normalizer. Descriptions regarding the normalizers for SUV assessments were added to the discussion section (Page 16, line 261 to 269). The additional results for SUV(lbm)max and SUV(bsa)max are shown in S2 and S3 Figs. The relationship between liver FDG uptake and age has already been shown in the literature (Clin Nucl Med. 2013,38(6):422-5; Clin Imaging. 2010,34(5):348-50), but we checked the relationship of liver SUV(bw)max, SUV(lbm)max, and SUV(bsa)max with the patients’ age also in our study population. Similar to the results of the skin values, liver SUVmax values were significantly lower in the age group 0–9 years compared to those in adults. The age-related increase in SUVmax was, however, more gradual in females than males, and the decrease in SUVmax found in the skin of older women was not seen in their corresponding liver values. The age-related tendency was not different among SUVmax normalizers, suggesting that the liver SUVmax calculated by different metrics did not affect the quantitative assessment of differences among age groups.

4. Comments: the authors simply made as much of the raw data available as possible (e.g. a table of heights, weights, ages, SUVmeans, SUVmaxes etc. for each subject) and using VOIs of the same thickness for all subjects.

Response: According to your kind suggestion, we added supplemental tables (S4 and S5 Tables) containing the raw data of patients for the assessment of repeatability, regional distribution, and relationship with age. The tables also include the patients’ age, BMI, and additional SUV metrics (lbs and bsa).

5. Comments: the PET camera used contain silicon photomultipliers, but there’s no clear reason to think that these per se had any real influence on this study.

Response: Indeed, the manuscript did not assess or even discuss the contribution of silicon photomultipliers to the accuracy in skin uptake measurements. Therefore, we deleted the term “silicon multiplier” and “SiPM” from the manuscript.

Reviewer #2: 

1. Comments: Discuss the possibility that the observed results are due to backscatter from bony surfaces in face, skull, sternum (not present in abdomen, etc.).

Response: Thank you very much for your comment. PET attenuation correction on PET/MRI is performed by MR-based pseudo-CT, which does not induce CT-derived artifacts such as beam-hardening artifacts. In the attenuation correction (AC) map (μ map) generated from MR-based pseudo-CT, only four-tissue segmentations (air, fat, water, and soft-tissue) were utilized for the correction without bone components except for the skull (CT-atlas or MR-based bone segmentation) (J Nucl Med 2012; 53:796–804). As you mentioned, CT-AC does have a “backscatter” phenomenon close to the bones, but MR-AC does not induce this artifact due to the lack of a bone component. We added descriptions regarding this issue (Page 15, line 257 to page 16, line 260).

2. Comments: Discuss the possibility that the observed results may be in part explained by the lack of self shielding in the higher noted uptake areas.

Response: The issue you comment on is important and is carefully addressed in the methods section. High uptakes by organs other than the skin including the brain, salivary glands, muscles, liver, and urinary system were excluded from the analysis, as described in the Image analysis section (Page 9, line 143 to page 10, line 147). As a result, manual adjustments were necessary for 69 of the 298 examinations. Nevertheless, a “spillover” effect as mentioned by the reviewer cannot be completely resolved and is described as a limitation of the study.

3. Comments: Discuss the possibility that the observed results may be the result of uptake in the blood pool (higher capillary blood at the surface in the noted higher uptake regions), in the eyes themselves (or salivary glands), in the nasopharangeal regions.

Response: Thank you for your valuable comments. We agree that the blood pool could be a possible reason for skin uptake. The SUVmax in the face, especially in the nose, was higher than that in the large vessels or mucosa of the nose (data not shown), and even higher than that in the blood-rich organ liver in 58% (131 of 224 examinations) of the patients (S5 Table) in our study. We consider that these results suggest that the skin uptake cannot be explained by the blood pool alone. The descriptions regarding the abovementioned results were added to the discussion section (Page 16, line 275 to page 17, line 279). As for the physiological uptakes of the orbit (external ocular muscle and lachrymal gland) and the tonsilla or salivary gland in the nasopharyngeal areas, we carefully exclude all those uptakes from the skin VOIs as shown in the Image analysis section. We add descriptions regarding those regions to the Image analysis section　(Page 10, line 145).

In addition to the revision of the manuscript according to the reviewers’ valuable comments, we corrected minor typographical errors as follows:

Table 1 (Page 6): Heading for the table.

Table 1 (Page 6): The interquartile range of the BMI of 1st examination for assessment of repeatability and regional distribution.

Table 1 (Page 6): The mean value of the BMI of female patients in 0-9 yrs. old for assessment for relationship with age.

Table 1 (Page 6): The interquartile range of the BMI of female patients in 0-9 yrs. old and 70-19 yrs. old for assessment for relationship with age.

Page 10, line 162: The name of the statistical analysis (paired t-test > the Wilcoxon’s signed-rank test).

Page 12, line 187-188: The percent mean difference and lower and upper limits of agreements of the SUVmax (and 95% CI) in the overall skin region.

Page 12, line 193-194: The coefficient of repeatability for the SUVmax (95% CI) in the overall skin region.

Page 14, line 227: The SUVmax of the face in female patients.

---

## [Decision Letter · Decision Letter 1]

22 Feb 2021

PONE-D-20-29259R1

Physiological skin FDG uptake: A quantitative and regional distribution assessment using PET/MRI

PLOS ONE

Dear Dr. Nogami,

Thank you for submitting your manuscript to PLOS ONE. After careful consideration, we feel that it has merit but does not fully meet PLOS ONE’s publication criteria as it currently stands. Therefore, we invite you to submit a revised version of the manuscript that addresses the points raised during the review process.

We look forward to receiving your revised manuscript.

Kind regards,

Matteo Bauckneht, MD, PhD

Academic Editor

PLOS ONE

Reviewers' comments:

Reviewer's Responses to Questions

**Comments to the Author**

1. If the authors have adequately addressed your comments raised in a previous round of review and you feel that this manuscript is now acceptable for publication, you may indicate that here to bypass the “Comments to the Author” section, enter your conflict of interest statement in the “Confidential to Editor” section, and submit your "Accept" recommendation.

Reviewer #1: (No Response)

Reviewer #2: All comments have been addressed

2. Is the manuscript technically sound, and do the data support the conclusions?

Reviewer #1: No

Reviewer #2: Yes

3. Has the statistical analysis been performed appropriately and rigorously? 

Reviewer #1: Yes

Reviewer #2: Yes

4. Have the authors made all data underlying the findings in their manuscript fully available?

Reviewer #1: Yes

Reviewer #2: Yes

5. Is the manuscript presented in an intelligible fashion and written in standard English?

Reviewer #1: Yes

Reviewer #2: Yes

6. Review Comments to the Author

Reviewer #1: While the manuscript is much improved by the additional data included within the revision, the similarity in the trends for the SUV body-weight values as a function of age between the liver and skin actually heighten my concern about statements and data within the manuscript that might be interpreted by the reader to conclude that there exists a trend related to absolute FDG uptake. Although it was good to include the additional types of SUV metric (lean body mass and body surface area), it is wrong to conclude that similarity in their trends as a function of age (for example) means that trend exists for absolute FDG uptake. It is quite possible that all of these body-habitus metrics are poor (i.e. biased) normalizers for pediatric subjects. And because of the ease with which this point can be missed, it is imperative that the authors make this distinction clear, including in the Abstract, which may well be as far as some readers get.

So, while it is acceptable to report results showing that SUV is correlated with age, the authors should take pains to note that these trends do not necessarily hold for “FDG uptake” because SUV has not been definitively shown to correlate with FDG uptake across this patient population. If I am mistaken about this, the authors should of course make the case and cite the evidence that refutes my position, but otherwise all language suggesting that SUV is a quantitative measure of FDG uptake should be removed. As a case in point, the new language on page 16 line 261 in the Discussion, I believe is misleading. SUV is neither a robust nor particularly quantitative measure of FDG uptake, and to suggest that “biological and technical factors” that might be confounding that relationship have been ruled out, is simply incorrect.

Reviewer #2: (No Response)

7. PLOS authors have the option to publish the peer review history of their article (what does this mean?). If published, this will include your full peer review and any attached files.

Reviewer #1: No

Reviewer #2: No

---

## [Author Response · Author response to Decision Letter 1]

3 Mar 2021

We thank the reviewers once again for evaluating of our study, and the editor for giving us the opportunity to resubmit our revised manuscript. We are happy to address all comments in a point-by-point manner. The revisions have been marked in colored font for ease of identification in the revised version of our manuscript.

Reviewer #1: 

1. Comments: While the manuscript is much improved by the additional data included within the revision, the similarity in the trends for the SUV body-weight values as a function of age between the liver and skin actually heighten my concern about statements and data within the manuscript that might be interpreted by the reader to conclude that there exists a trend related to absolute FDG uptake. Although it was good to include the additional types of SUV metric (lean body mass and body surface area), it is wrong to conclude that similarity in their trends as a function of age (for example) means that trend exists for absolute FDG uptake. It is quite possible that all of these body-habitus metrics are poor (i.e. biased) normalizers for pediatric subjects. And because of the ease with which this point can be missed, it is imperative that the authors make this distinction clear, including in the Abstract, which may well be as far as some readers get.

Response: Thank you very much for your valuable comments. We completely agree with your opinion that the measured SUV could be highly related to the biased normalizer for body habitus, especially in pediatrics. Indeed, the SUV in the liver was also lower in pediatrics than in adult patients of both sexes, which suggests, as you mentioned, that the difference in skin SUV in pediatrics was not related to absolute FDG uptake at their age, but rather the biased normalization procedure for SUV based on their body habitus. We also agree that this influence of body size on accurate measurement of tracer uptake is a significant drawback of the SUV, which has already been confirmed more than twenty years ago (J Nucl Med 1995; 36:1836-1839. SUV: Standard Uptake of Silly Useless Value?). According to the reviewer’s valuable comments and the literature showing the limitations of SUV due to differences in body size, we have revised the discussion section’s description of the relationship between SUV and age groups (Page 16, lines 262 to 276). In addition, the conclusion sections in the Abstract and the main text have been revised accordingly.

2. Comments: So, while it is acceptable to report results showing that SUV is correlated with age, the authors should take pains to note that these trends do not necessarily hold for “FDG uptake” because SUV has not been definitively shown to correlate with FDG uptake across this patient population. If I am mistaken about this, the authors should of course make the case and cite the evidence that refutes my position, but otherwise all language suggesting that SUV is a quantitative measure of FDG uptake should be removed. As a case in point, the new language on page 16 line 261 in the Discussion, I believe is misleading. SUV is neither a robust nor particularly quantitative measure of FDG uptake, and to suggest that “biological and technical factors” that might be confounding that relationship have been ruled out, is simply incorrect.

Response: Thank you again for your valuable suggestions. We agree that the trends observed in the pediatrics in this study were not due to differences in FDG uptake, but rather differences in body size. Accordingly, we have removed the incorrect descriptions regarding “robustness” and “ruling out the biological and technical factors” from the discussion section (Page 16, line 262 and line 265). 

In addition to the aforementioned revisions made according to the reviewers’ valuable comments, we have corrected a minor typographical error as follows:

Page 25, line 4: Figure legend for S4 Fig.

We thank the reviewers once again for their significant commitment in reviewing our manuscript and the kindness shown throughout this process.

---

## [Editor Report · Decision Letter 2]

16 Mar 2021

Physiological skin FDG uptake: A quantitative and regional distribution assessment using PET/MRI

PONE-D-20-29259R2

Dear Dr. Nogami,

We’re pleased to inform you that your manuscript has been judged scientifically suitable for publication and will be formally accepted for publication once it meets all outstanding technical requirements.

Kind regards,

Matteo Bauckneht

Academic Editor

PLOS ONE
---

## [Editor Report · Acceptance letter]

18 Mar 2021

PONE-D-20-29259R2 

Physiological skin FDG uptake: A quantitative and regional distribution assessment using PET/MRI 

Dear Dr. Nogami:

I'm pleased to inform you that your manuscript has been deemed suitable for publication in PLOS ONE. Congratulations! Your manuscript is now with our production department. 

Kind regards, 

on behalf of

Dr. Matteo Bauckneht 

Academic Editor

PLOS ONE